# Development of Hydrogen-Permselective Porous Membranes Using Radiation-Induced Graft Polymerization

**Shin Hasegawa [1],\***, **Shinichi Sawada [1]**, **Shinya Azami [2]**, **Tokio Hagiwara [2]**, **Akihiro Hiroki [1]** **and Yasunari Maekawa [1],\***

[1] Functional Polymer Research Project, Quantum Beam Science Research Directorate, Takasaki Advanced Radiation Research Institute (TARRI), National Institutes for Quantum and Radiological Science and Technology (QST), 1233 Watanuki, Takasaki, Gunma 370-1292, Japan; sawada.shinnichi@qst.go.jp (S.S.); hiroki.akihiro@qst.go.jp (A.H.)

[2] Department of Environmental Engineering, Saitama Institute of Technology (STI), 1690 Okabe-cho, Osato-gun, Saitama 369-0293, Japan; setsuori48ggo@yahoo.co.jp (S.A.); setsuori34@yahoo.co.jp (T.H.)

\* Correspondence: hasegawa.shin@qst.go.jp (S.H.); maekawa.yasunari@qst.go.jp (Y.M.); Tel.: +81-27-346-9410 (Y.M.)

**Abstract:** Hydrogen-permselective membranes were developed using a radiation-induced grafting method. Styrene (St) and acrylic acid (AAc) monomers were introduced into porous polyvinylidene fluoride (PVDF) membranes to obtain St- and AAc-grafted PVDF membranes with grafting degrees of 82% and 92%, respectively. The porosities of the grafted membranes were controlled in the range 30–40% by hot-press compression at 159 °C and 4 MPa. The hydrogen permeability was found to be of the order of $10^{-7}$ mol/m$^2$·s·Pa, which was higher than the permeability for water vapor and nitrogen (oxygen model). The St- and AAc-grafted membranes exhibited 9.0 and 34 times higher permeability for $H_2$ than for $H_2O$ and $N_2$, respectively.

**Keywords:** hydrogen permselective membrane; radiation-induced graft polymerization; porous membrane

## 1. Introduction

In 2011, hydrogen explosions in the reactor buildings at Fukushima Daiichi nuclear power station caused widespread radioactive contamination of the environment [1]. There are two main ways to avoid hydrogen explosions in reactor buildings in the event of a nuclear accident involving a loss of external power. One of these is the release of hydrogen to the atmosphere by installing a ventilation system with a filter, and the other is to recover hydrogen using collection equipment [2]. The former system may cause the release of radioactive contaminants into the atmosphere, as it is difficult to completely remove radioactive pollutants. In the latter system, the hydrogen collection equipment is composed of a recombiner using powered equipment containing hydrogen storage alloys consisting of many materials such as platinum and rare earth elements. In the nuclear accident at Fukushima, these devices did not work due to the loss of power. Therefore, the main requirements of such a system are "to operate hydrogen collection equipment passively without a power source" and to use "resources with realistic prices" [3].

To realize the passive hydrogen storage system described above, a magnesium alloy, developed by Hashimoto et al., is one of the most promising target materials [4]. Since magnesium has a risk of burning through contact with moisture, a radiation-resistant hydrogen-permselective membrane must be developed in order to prevent moisture from reaching the alloy. Therefore, in this study,

we developed a hydrogen-permselective membrane that selectively allows hydrogen to permeate at a moderate rate while simultaneously blocking oxygen and water vapor.

In previous reports, ceramic thin films have been proposed as a commercially available hydrogen-permselective membrane for high-temperature (>500 °C) operation [5,6]. Polyimide films have also been developed as hydrogen-permselective membranes for low-temperature environments (<100 °C) [7]. These membranes have high hydrogen permselectivity to oxygen and nitrogen gases, together with high water vapor permeability. However, water vapor barrier films currently on the market also decrease the permeability to most gases, including hydrogen, in proportion to their barrier property ($10^{-15}$ mol/m$^2$·s·Pa) [8].

There are two methods for producing a film having a high-enough hydrogen permeation rate while also having permselectivity to other gases including water vapor. A trade-off in properties can be achieved by taking advantage of the "molecular sieve effect" and the "interacting effects between gas molecules and the polymer membrane structure." The molecular sieve effect uses differences in the size of gas molecules to selectively allow the permeation of hydrogen (molecular size: $H_2 \leq H_2O < O_2$). A gas-permselective membrane can be made such that it will selectively allow the permeation of hydrogen (or water vapor) but not oxygen, which has a larger molecular size, by controlling the size of the pores.

The radiation-induced graft polymerization technique has the ability to introduce new functional graft polymers into solid polymer substrates while maintaining their mechanical properties. There have been many reports concerning membrane durability in severe environments for applications including fuel cells, water desalination, and precious or hazardous metal recovery from environmental and industrial wastewater [9–14]. Pore-filling membrane is also a candidate used in the development of anion exchange membrane (AEM) with excellent chemical and mechanical strength. These membranes are usually prepared by filling a porous substrate such as PE, PP, or PTFE with a polymer electrolyte. This concept was proposed by Yamaguchi et al., which could simultaneously complement the mechanical and chemical stability of ion-exchange membranes [15,16]. In this study, we applied the graft polymerization technique to control the molecular sieve effect and the interaction effects of gas molecules interacting with the graft polymer chains (Scheme 1).

**Scheme 1.** Radiation induced-graft polymerization of AAc and St into porous PVDF membrane.

A hydrogen-permselective membrane requires a moderate permeation rate ($10^{-8}$–$10^{-9}$ mol/m$^2$·s·Pa) for the supply of the gas to the recombiner to prevent hydrogen explosions in nuclear power plants. Furthermore, the recombiner systems and materials are exposed to moisture at elevated temperature (>100 °C). From the above requirements, we first ruled out non-porous membranes, which have lower permeability than $10^{-11}$ mol/m$^2$·s·Pa. Thus, we selected porous PVDF membranes, which possess thermal stability and water resistance due to hydrophobicity. Since the permeability of porous PVDF is too high (>$10^{-5}$) to induce the permselectivity of $H_2$ over $H_2O$ and $O_2$, which should be blocked out. Thus, for a molecular sieve effect, the porosity of the porous PVDF membranes can be controlled by the quantity of introduced graft polymers [17–21]. For an interaction effect, we employed polystyrene (Poly-St) and polyacrylic acid (Poly-AAc) as hydrophobic and hydrophilic graft polymer chains, respectively, to control the solubility and the interaction with water vapor. Since Poly-St and Poly-AAc

have a hydrophobic phenylene group and hydrophilic carboxyl groups, respectively, those functional groups are considered to block water vapor by hydrophobic repulsion and to reduce the permeability of water molecules by hydrophilic interactions. We examined the type and the number of graft polymers to control the hydrophilicity/hydrophobicity and porosity to find the appropriate conditions for a high hydrogen diffusion rate and hydrogen permselectivity against water vapor. In addition, to control gas permeability, the porous PVDF membrane and porous graft membrane were heated and compressed (hot pressed) to control their porosity and to achieve the abovementioned permeation rate with permselectivity of hydrogen.

## 2. Materials and Methods

Polyvinylidene fluoride (Durapore Membrane Filter GVWP09050) with a 0.22 μm average pore size was purchased from Merck Millipore and used without treatment. Acrylic acid (AAc, 98.0%), styrene (St, 99.0%), toluene (99.5%), ethanol (99.5%), and ammonium iron (II) sulfate, so-called Mohr's salt (99.0%), were purchased from Wako Pure Chemical Industries, Ltd., Osaka, Japan, and were used without further purification. Deionized water was purified using the Millipore Milli-Q UV system to produce a resistivity of 18.2 MΩ cm and a total organic carbon content of <10 ppb.

### 2.1. Sample Synthesis

Porous PVDF membranes were irradiated with doses ranging from 30 to 160 kGy (dose rate: 10 kGy/h) using γ-rays from a $^{60}$Co source (QST, Takasaki, Gunma, Japan) under an argon atmosphere at room temperature. Samples were then immersed in an AAc/$H_2$O or St/toluene monomer solution under an argon gas atmosphere to initiate the grafting. The graft polymerizations of the AAc and St monomers were carried out at 30 °C, 50 °C, or 60 °C for 0.1–7 h. For AAc, 0.05 wt% ammonium iron (II) sulfate was added to avoid homopolymerization [22,23]. After graft polymerization, all AAc- and St-grafted samples were washed with excess amounts of acetone and toluene several times to remove homopolymers and monomers. They were then dried under vacuum at 40 °C for 12 h. We calculated the grafting degree (GD) using the following equation:

$$GD(\%) = 100 \times \frac{W_1 - W_0}{W_0} \tag{1}$$

where $W_1$ and $W_0$ are the membrane weights before and after graft polymerization, respectively.

### 2.2. Heat-Compression Molding

The grafted membrane was cut into an 8 cm diameter disk, and sandwiched with grid sheet (0.12 mm thickness APICA Co. Ltd., Tokyo, Japan) of 9 cm disk in a diameter and insert into two glass plates with 11 cm × 11 cm square. Heat-compression molding of these glass plate sets was carried out using IMC-16FE (Imoto Machinery Co., Ltd., Kyoto, Japan) under 4 MPa at 159 °C for 1 min.

### 2.3. Porosity

The porosity is a parameter that describes the volume fraction of voids in the membrane, as calculated from the below equation:

$$\text{Porosity } (\%) = 100 - 100 \times \frac{V_2 + V_1}{V_0} \tag{2}$$

where $V_0$, $V_1$, and $V_2$ are the volume of the grafted porous membrane, the volume occupied by the base material (PVDF), and the volume occupied by the grafted chain, respectively. The volume of the film was calculated from the ratio of the film thickness multiplied by the film surface area (vertical and horizontal length) before and after the reaction.

## 2.4. Gas Permeation Test

Gas permeation tests of grafted porous membranes were conducted by a vacuum method according to the previous report [24]. The grafted samples were cut into disks (10 mm in a diameter). The effective permeation area of the membrane was estimated to be 0.07 cm$^2$. The Experimental setup is presented schematically in Scheme 2. The membranes were sandwiched and supported by stainless-steel meshes (100 mesh) to avoid changes in the effective area of the membrane resulting from deformation caused by the vacuum. Hydrogen, nitrogen, and water vapor were supplied alternately at atmosphere pressure on the gas-feed side of the membrane. The chamber was flushed with an individual gas for at least 15 min at 5 mL/min gas flow. Water vapor flow with N$_2$ gas at 95 °C was maintained for 45 min at 5 mL/min. After stopping the gas feed, the pressure change in the stainless-steel chamber was recorded. The gas permeability was calculated using the following equation:

$$\text{Gas Permeability} \left( \text{mol}/ \sec \cdot \text{m}^2 \cdot \text{s} \cdot \text{Pa} \right) = \frac{\Delta \text{Pa} \times V/RT}{x \times A \times P} \tag{3}$$

where ΔPa is the pressure difference before and after the permeability experiment and *V*, *R*, *T*, *x*, *A*, and *P* are the volume of the chamber (0.05 L), the gas constant, the temperature, the measuring time, the effective area of the membrane, and the atmospheric pressure, respectively. The permselectivity of hydrogen gas against water vapor and nitrogen was evaluated from the permeability ratios of hydrogen (H$_2$) with water vapor (H$_2$O) and nitrogen (N$_2$) and are defined as R(H$_2$/N$_2$) and R(H$_2$/H$_2$O), respectively.

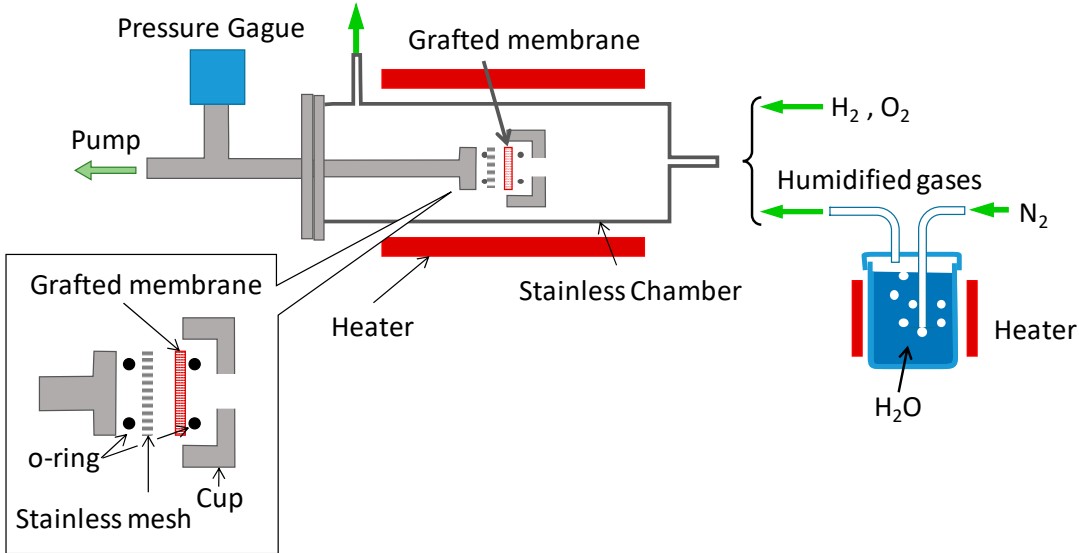

**Scheme 2.** Schematic illustration of gas permeation experiments.

## 3. Results and Discussion

### 3.1. Synthesis

The AAc and St graft polymerization of the porous PVDF membranes (AAc-grafted PVDF and St-grafted PVDF, respectively) were performed with different irradiation and grafting conditions to obtain GDs in the range 0–100%. In the case of AAc, the porous PVDF membranes, pre-irradiated with a dose of 160 kGy, were immersed in an AAc/H$_2$O (10:90 *v/v*) solution at 30 °C for 0.1–1 h with 0.05 wt% Mohr's salt to suppress homopolymer formation. By changing the AAc concentration in the range 10–70% in the monomer solution, the GDs could be controlled in the range 25–92%. In the case of St graft polymerization, the PVDF membranes, pre-irradiated with a dose of 30 kGy, were immersed in St/toluene (50:50 or 90:10 vol%) at 50 °C or 60 °C for 1–7 h to obtain St-grafted

PVDF with GDs ranging from 11% to 82% (Table 1). St-grafted PVDF have different GDs (63% and 82%) with same reaction condition. In general, radiation-induced graft polymerization gives GDs with a distribution of about ±10–20% even under the same reaction condition because of radical inactivation by the oxygen concentration in the monomer solution. However, the membranes having different GDs obtained in the same reaction conditions were utilized the permeation experiment to compare hydrogen permselectivity of the membranes with different GDs [25]. Since the gas permeation performance is considered to depend largely on the amounts of introduced graft polymers (i.e., GDs), we first focus on the effects of GD by preparing the St-grafted PVDF having different GDs (63% and 82%) with same reaction condition even though the permselectivity of the membranes should be affected to some extent by the structures of graft polymers, such as length and the number of graft polymer chains, which are affected by absorbed doses and the monomer concentrations in a graft polymerization process. The porosity was calculated from the size, thickness, and weight of the membrane before and after the graft polymerization. Despite the change in GD, the porosities of the grafted membranes remained around 60%, which was almost the same as the substrate porosity (61%). In the gas permeation tests using the substrate, the permeability was $10^{-6}$ mol/m$^2$·s·Pa, which is too high to exhibit preferential hydrogen permeation. This is because membranes with a porosity of around 60% possess large voids with a size scale of more than 1 μm, which are too large to have a molecular sieve effect or an affinity effect with a graft chain.

**Table 1.** Synthetic results of St- and AAc-grafted PVDF by radiation-induced graft polymerization.

| Graft Chains | Monomer Solutions | Dose (kGy) | Reaction Temp (°C) | Reaction Time (h) | GD (%) | Porosity (%) | Porosity after Heat Compression (%) [*1] |
|---|---|---|---|---|---|---|---|
| Substrate (PVDF) | - | - | - | - | 0 | 61 | - |
| Poly (AAc) | AAc/H$_2$O (10:90 wt%) [*2] | 160 | 30 | 1 | 25 | 60 | 36 |
| | AAc/H$_2$O (20:80 wt%) [*2] | 160 | 30 | 1 | 63 | 61 | 26 |
| | AAc/H$_2$O (70:30 wt%) [*2] | 160 | 30 | 0.1 | 76 | 61 | 39 |
| | AAc/H$_2$O (70:30 wt%) [*2] | 160 | 30 | 0.2 | 92 | 61 | 36 |
| Poly (St) | St/Toluene (50:50 wt%) | 30 | 60 | 1 | 11 | 60 | 48 |
| | St/Toluene (50:50 wt%) | 30 | 60 | 7 | 20 | 58 | 44 |
| | St/Toluene (90:10 wt%) | 30 | 50 | 1 | 43 | 53 | 33 |
| | St/Toluene (50:50 wt%) | 30 | 60 | 4 | 63 | 54 | 44 |
| | St/Toluene (50:50 wt%) | 30 | 60 | 4 | 82 | 54 | 44 |

[*1] 159 °C, 4 MPa, 10 min, [*2] 0.05 wt% Mohr's salt.

The hot-press compression treatment of the grafted porous PVDF membranes reduces the porosity, i.e., the pore diameter, resulting in an increase of the contact time between the gas molecules and the pore surfaces in the membranes. Since the permeability of the membranes should be of the order of $10^{-9}$ mol/m$^2$·s·Pa or higher for practical use in hydrogen separation, we tried to optimize the hot-press conditions, such as temperature and pressure, for the grafted porous PVDF substrates so that they had a permeability of approximately $10^{-9}$ mol/m$^2$·s·Pa. We found that the optimal conditions were 159 °C at 4 MPa for 10 min (see Supplementary Materials, Table S1). The H$_2$ permeability of the grafted membranes decreased to a level in the range 26%–44% (Table 1). After the hot-press compression treatment, each grafted membrane showed no weight loss, because the poly(AAc) and poly(St) graft chains have decomposition temperatures of 200 °C and 325 °C, respectively [26,27].

As shown in Figure 1a, the untreated porous PVDF substrate contains three-dimensional network structures with pore diameters of approximately 1 μm or less. As shown in Figure 1b, the graft

polymerization of St makes the porous PVDF membranes expand by the amount of St that was incorporated in the membrane while keeping the shape and size of pores present in the original substrate. This is not our expected result for the strategy that aimed to reduce the pore size by filling them with graft polymers by radiation-induced grafting [28,29]. However, from another perspective, this means it is possible to introduce new functional graft polymers into the porous substrates while retaining their overall porosity and pore size. The heat-compression treatment of the St-grafted PVDF decreased the number and the size of the pores, as shown in Figure 1c, resulting in scattered pores with a diameter of about 1 μm.

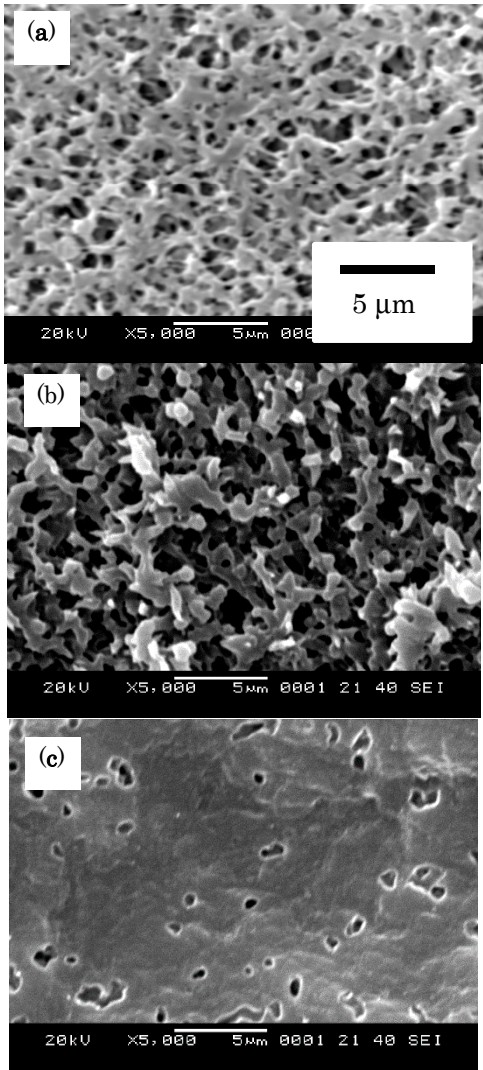

**Figure 1.** SEM micrographs of surface of St grafted porous PVDF membrane: (**a**) PVDF substrate membrane, (**b**) St-grafted PVDF (GD 96%), (**c**) St-grafted PVDF (after heat compression treatment), respectively.

### 3.2. Gas Permeation Property

To adjust the permeability in the range $10^{-9}$–$10^{-8}$ mol/m$^2$·s·Pa as described in the previous section, membranes having a porosity of 28%–38% were utilized to examine the effect of the GD on the gas permeability. The gas permeability of $H_2$, $N_2$ ($O_2$ model), and $H_2O$ and the permeation ratios of $H_2$ against $N_2$ ($R(H_2/N_2)$) and against $H_2O$ ($R(H_2/H_2O)$) through the St- and AAc-grafted membranes are listed as an index of $H_2$ permselectivity in Table 2. The values for $H_2$ and $H_2O$ as a function of GD are highlighted in Figure 2.

**Table 2.** $H_2/N_2$($O_2$ model) and $H_2/H_2O$ ratios through the St- and AAc-grafted PVDF membranes.

| Grafted Chains | GD (%) | Porosity (%) | Gas Permeability ($\times10^{-8}$ mol/sec·m²·Pa) | | | R($H_2/N_2$) | R($H_2/H_2O$) |
|---|---|---|---|---|---|---|---|
| | | | $H_2$ | $N_2$ | $H_2O$ | | |
| AAc | 25 | 36 | 41 | 20 | 18 | 2.1 | 2.3 |
| | 63 | 26 | 32 | 15 | 5.2 | 2.1 | 6.2 |
| | 76 | 39 | 9.4 | 3.2 | 2.9 | 2.9 | 3.2 |
| | 92 | 36 | 6.8 | 2.3 | 0.2 | 3.0 | 34 |
| St | 11 | 48 | 5.6 | 3.7 | 1.5 | 2.5 | 3.7 |
| | 20 | 44 | 1.8 | 0.55 | 0.6 | 3.3 | 3.0 |
| | 43 | 33 | 0.6 | 0.13 | 0.1 | 4.6 | 6.0 |
| | 63 | 44 | 0.3 | 0.14 | 0.09 | 2.1 | 3.0 |
| | 82 | 44 | 0.03 | 0.40 | 0.003 | 2.5 | 9.0 |

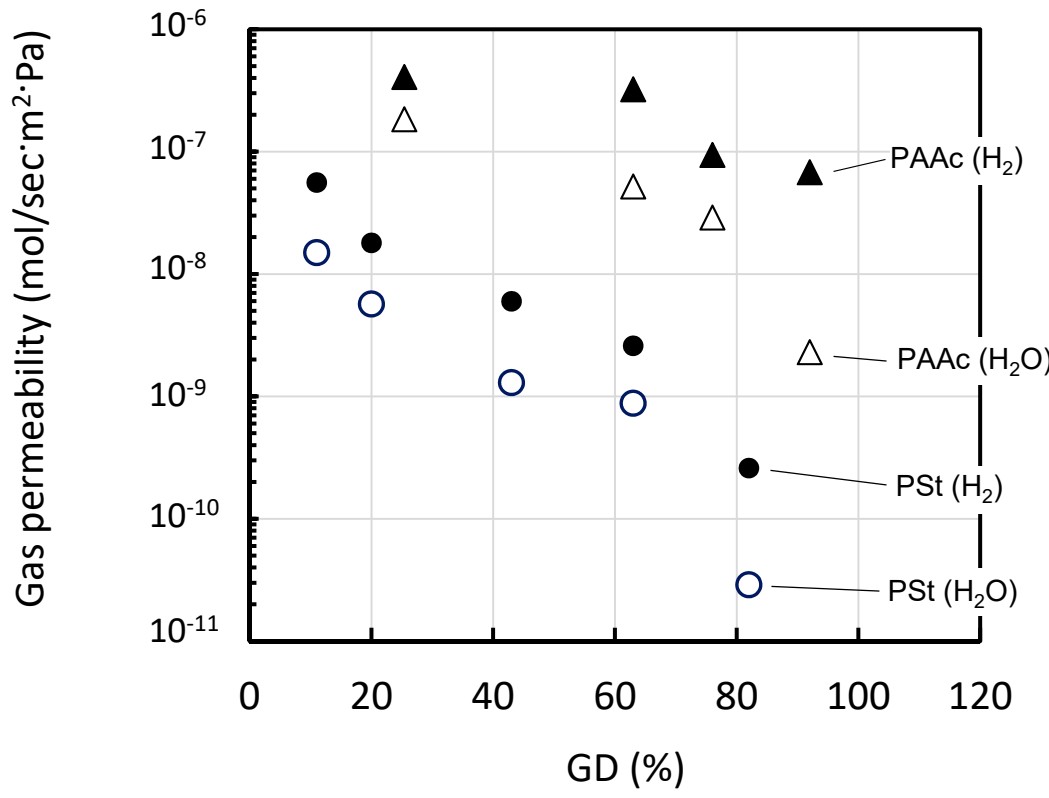

**Figure 2.** The gas permeability (mol/sec·m²·Pa) of $H_2$ and $H_2O$ as a function of GD at 95 °C. AAc-grafted PVDF: (▲) $H_2$, (△) $H_2O$. St grafted PVDF: (●) $H_2$, (○) $H_2O$.

The permeability of the AAc-grafted PVDF decreased from $10^{-7}$ to $10^{-8}$ mol/m²·s·Pa with an increase in the GD from 25% to 92%. The gas permeability of the St-grafted membranes decreased more steeply (from $10^{-7}$ to $10^{-11}$ mol/m²·s·Pa) as the GD increased from 11% to 82%. These results indicate that the introduced graft polymers reduced the pore sizes in the membranes, even though it was not possible to see the GD dependence of the porosity in either type of grafted membrane. As shown in Figure 3, AAc-grafted membranes also keep the shape and size of pores alike those of St-grafted membranes, the St-grafted membranes showed lower permeability than the AAc-grafted membranes. This is probably because the hydrophobic St-grafted chains are more miscible with the hydrophobic PVDF substrates, resulting in a decrease in the pore size or connectivity through the membrane. As shown in Figure 1c or Figure 3b, the pore sizes of the graft membrane are significantly

reduced by hot pressing, lowering the permeability by the three order of magnitude (from $<10^{-5}$ to $10^{-8}$ mol/ sec·m$^2$·s·Pa sec). Even though, changes in the pore sizes are not clear, the numbers of pore appear to be reduced, judging from the SEM images. Therefore, the gas molecules permeate through the holes should collide more with the pore wall and also through the polymer balk region in Figure 1c or Figure 3b.

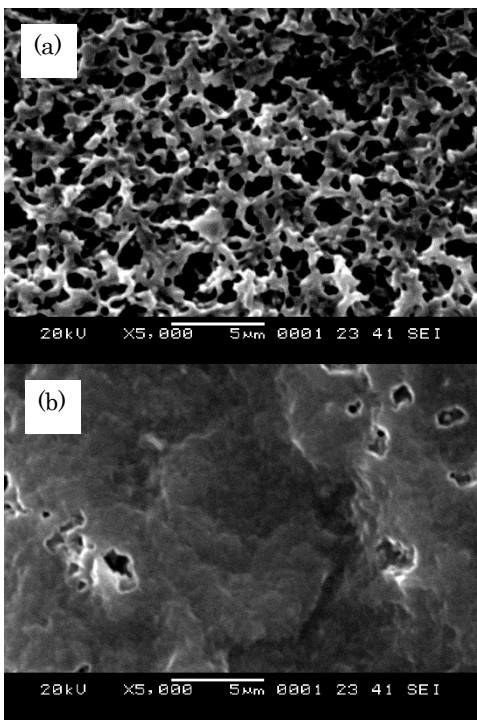

**Figure 3.** SEM micrographs of surface of AAc-grafted PVDF: (**a**) AAc-grafted PVDF (GD 25%) and (**b**) AAc-grafted PVDF (after heat compression treatment), respectively.

Both St- and AAc-grafted membranes showed distinguishable $H_2$ permselectivity against $N_2$ ($R(H_2/N_2)$), with values in the range 2.1–4.6 regardless of the GD. The permselectivity of these membranes against $N_2$ seemed not to be affected by the amount the graft polymer that was incorporated. However, the $H_2$ permselectivity ratio to $H_2O$ ($R(H_2/H_2O)$) values was higher and exhibited a clear GD dependence. Thus, at low GD (<20%), both grafted membranes showed similar $H_2$ permselectivity ratios against $H_2O$ ($R(H_2/H_2O)$) compared with those against $N_2$ ($R(H_2/N_2)$), whereas the $R(H_2/H_2O)$ values in both membranes increased from 2–3 to approximately 10 or more at higher GDs (>80%). These results clearly demonstrate that regardless of whether the introduced graft polymer is hydrophobic (St) or hydrophilic (AAc), a certain amount of graft polymer improves the $H_2$ permselectivity against $H_2O$ by approximately one order of magnitude, but the permselectivity against $N_2$ does not change. This is probably because the introduced graft polymers alter the size and connectivity of the pores even when there is the same overall level of porosity. The pore shape in the AAc-grafted membranes was also retained, as in the St-grafted membranes. Furthermore, the compression treatment reduced the number of holes in the AAc membranes, as was the case for the St-grafted membranes.

### 3.3. Structure and Mechanical Properties

The distribution of grafted chains in AAc-grafted PVDF before and after the compression treatment in the cross-sectional direction was observed through scanning electron microscopy/energy-dispersive X-ray spectroscopy, as shown in Figure 4a,b, respectively. The AAc-grafted membranes were stained with Cu ions by an ion exchange reaction to observe the distribution of the acrylic acid of the poly (AAc) graft polymers in the cross-sectional direction of the dried sample. In Figure 4, the yellow line indicates

the location of the elemental analysis of the membrane, and the red line indicates the intensity of Cu ions coordinated to -COO⁻. It is clear that the AAc-grafted polymers were successfully introduced homogeneously from the surface to the inner part of the membranes in both the untreated and the corresponding compression-treated membranes by the optimized graft polymerization conditions. Although it is difficult to judge the distribution of the cross section of the styrene graft chain in the St-grafted membrane, it is considered that the graft chains are uniformly distributed in the cross section from the improvement of the selectivity of the gas permeation test.

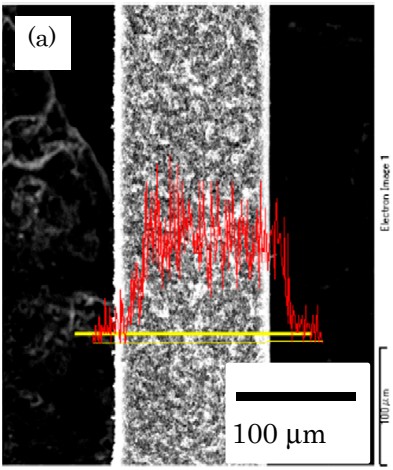

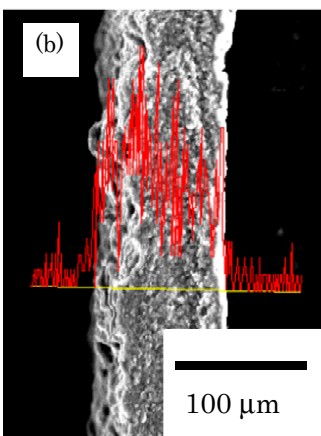

**Figure 4.** SEM micrographs of EDS analysis of cupper in the perpendicular direction of AAc-graft-PVDF: (**a**) before and (**b**) after compression treatment, respectively.

One of the factors that are required for putting synthesized membranes to practical industrial use is mechanical strength. Tensile strength and elongation at break are the most common indices for this mechanical property, because tearing and flexibility of the sample are factors that relate to durability, especially in membrane permeation systems. Figure 5 shows the tensile strength and elongation at break of the grafted samples plotted as a function of GD. The tensile strength and elongation of the original porous PVDF substrate were 10 MPa and 16%, respectively. After hot-compression treatment, the tensile strength of the substrate increased by 2.1 times (from 10 MPa to 21 MPa), while the elongation decreased to two-thirds of its original value (from 16% to 10%). The tensile strength of the AAc- and St-grafted membranes increased with increasing GD. Specifically, the AAc- and St-grafted membranes having GDs of 76% and 87% showed 47 MPa and 37 MPa, respectively. The elongation decreased drastically after the hot-press compression process, even though the grafted membranes possess certain

amounts of graft polymers. Almost no dependence of the elongation at break on GD was observed. All grafted samples were strong enough and flexible enough to prepare them for the gas permeation experiments to evaluate their gas permeability properties.

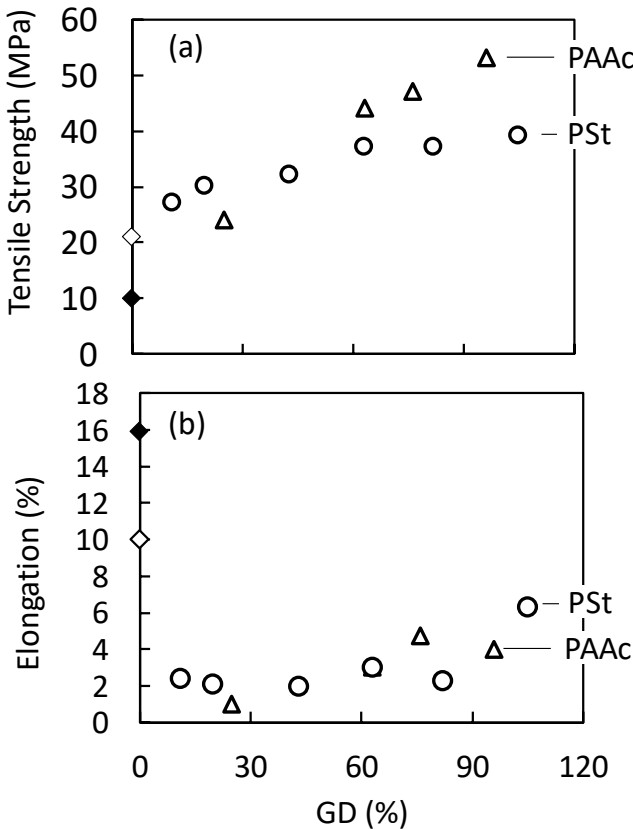

**Figure 5.** Tensile strength (**a**) and elongation at break (**b**) of porous PVDF membrane and grafted porous PVDF membranes before and after heat compression as a function of GD: PVDF membrane substrate before (◆) and after (◇) heat compression, and (○) St, (△) AAc-grafted PVDF after heat compression.

## 4. Conclusions

Hydrogen-permselective membranes were developed using a radiation-induced grafting method. Styrene and acrylic acid monomers were introduced into porous PVDF membranes to obtain St- and AAc-grafted PVDF with GDs of 82% and 92%, respectively. The porosities of the graft-polymerized membranes were maintained at around 60% before and after the graft polymerization. The permeability of the grafted membranes was maintained in the range $10^{-6}$–$10^{-7}$ mol/m$^2$·s·Pa, which was the same as the permeability of the base membrane. A decrease in porosity was obtained by hot-press compression treatment to reduce the diameters of the pores, resulting in an increase in the contact time between the gas molecules and the pore surfaces. The hydrogen permeability through the grafted membrane after the hot-press treatment was of the order of $10^{-7}$ mol/m$^2$·s·Pa and was higher than the permeability of water vapor or nitrogen (oxygen model). The St- and AAc-grafted membranes exhibited permeability for H$_2$ that was 9.0 and 34 times higher than the permeability for H$_2$O and N$_2$, respectively. The tensile strength of the AAc- and St-grafted membranes increased with increasing GD. The AAc- and St-grafted membranes showed tensile strengths of 47 MPa at a GD of 76% and 37 MPa at a GD of 87%, respectively. All grafted samples were strong and flexible enough to prepare them for the gas permeation experiments to evaluate their gas permeability properties.

**Supplementary Materials:** The following are available online at http://www.mdpi.com/2412-382X/4/2/23/s1, Table S1: Air permeability of porous PVDF film after hot press treatment.

**Author Contributions:** Conceptualization, Y.M.; methodology, S.H. and Y.M.; validation, S.H. and Y.M.; formal analysis, X.X.; investigation, S.A. and S.S.; Methodology, S.H.; Project administration, Y.M.; resources, S.A. data curation, S.H.; writing—original draft preparation, S.H.; writing—review and editing, Y.M., A.H.; visualization, S.H.; supervision, Y.M.; project administration, Y.M.; funding acquisition, T.H. and Y.M. All authors have read and agreed to the published version of the manuscript.

**Funding:** A part of this research was funded by JSPS KAKENHI Grant Number JP16K06785.

**Acknowledgments:** The authors would like to thank Enago for the English language review.

**Conflicts of Interest:** The authors declare no conflict of interest.

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
