# Peer review of "Development of Hydrogen-Permselective Porous Membranes Using Radiation-Induced Graft Polymerization"

_qubs, doi:10.3390/qubs4020023_

Round 1
Reviewer 1 Report
Dear the Authors,
This manuscript must be written clearly the methods adequately described.
Sincerely,
Reviewer 2 Report
The current manuscript reports on development of hydrogen-permselective membranes. Radiation-induced graft polymerization of St or AAc was applied to a PVDF porous membrane and hot-press compression was applied to the grafted membrane to decrease its porosity. The features of the obtained membrane were well characterized by SEM observation, gas permeability, SEM-EDS, and tensile tests. However, the manuscript lacks appropriate comparison with previous researches and rational strategy for the membrane development.
It hard to find the reasons from the manuscript why authors selected porous material as a substrate and selected radiation-induced graft polymerization to reduce the pore size. For the gas-permeation, probably non-porous materials are suitable for the substrate. For the pore size reduction, probably surface-initiated graft polymerizations are suitable, such as “pore-filling” research of Dr. Yamaguchi (Hollow-fiber-type pore-filling membranes made by plasma-graft polymerization for the removal of chlorinated organics from water, Journal of Membrane Science, Vol. 194, No. 2, pp. 217-228, 2001. ) Therefore, I recommend major revision. Authors should cite and compare with appropriate previous papers to clarify the novelty of this manuscript.
Below are the comments to the manuscript:
1) Authors should explain the reasons why they chose porous PVDF material as the substrate.
2) Authors should explain the reasons why they chose radiation-induced graft-polymerization. Why not surface-initiated graft polymerization? At least, solvent optimization seems to be needed for the radiation-induced graft polymerization.
3) Authors at least should compare and explain the advantage or difference with “pore-filling” research of Dr. Yamaguchi as mentioned above.
4) Why the dose was different from St and AAc ? Why 30 and 160 kGy?
5) Authors compared the membranes of almost same GD for the gas permeation property. The differences of the dose and monomer concentration does not need to consider?
6) Authors have to revise the sentence below (line 162-163) and cite appropriate papers in Introduction. Radiation-induced graft polymerization has been applied to the porous materials from 1980s or from earlier. The phenomenon of keeping the shape and pores after grafting was observed in several hundreds of papers.
“This is an unexpected result from a strategy that aimed to reduce the pore size by filling them with graft polymers by radiation-induced grafting”
7) What is the scientific ressons of the gas permeability difference between polySt- and polyAAc-grafted membranes?
8) In Figure 5, explanations of the meaning of the plots are missing. Explanation of x-axis is also missing. If the axis is same between (a) and (b), the graph should align. Plus, removal of the grid-lines maybe better.
Reviewer 3 Report
The article by Hasegawa et.al describes development and evaluation of the hydrogen-permselective membrane. The membrane is made using a radiation technique, which is interesting. Inexpensive and highly selective hydrogen permeable membranes are needed in various fields and are interesting. Although the subject is interesting, and perhaps the results are somewhat important, some of the figures and tables presented in this paper are incomplete in the following points and hard to understand. In my opinion it should be rejected.
1. Page 5, line 155. They described “The H2 permeability of the grafted membranes decreased to a level in the range 26–44% (Table 1).” I can not find the H2 permeability in fig. 1. Does it mean porosity after heat compression?
2. The description of Figure 3 is missing from the text.
3. I do not understand what each marker shape in Fig. 7 indicates, so I can not understand it. Also, what does the horizontal axis in (a) indicate?
4. The authors should indicate what the red and yellow lines in Figure 4 are. Why is the red line in (a) off the cross section?
5. Why is the GD quite different under the same conditions for the St / Tr (50:50 wt%) sample in Fig. 1?
6. Page 5, line 148. The authors claim “The hot-press compression treatment of the grafted porous PVDF films reduces the porosity, i.e., the pore diameter, resulting in an increase of the contact time between the gas molecules and the pore surfaces in the membranes”. Authors need to explain the mechanism. Why does the reduced pore diameter cause an increase of the contact time?
7. Page 8, line 186. The authors claim “These results indicate that the introduced graft polymers reduced the pore sizes in the membranes, even though it was not possible to see the GD dependence of the porosity in either type of grafted membrane.” They measured SEM image. They should evaluate it quantitatively from SEM images. From the SEM images of FIGS. 1 and 3, the pore diameter appears to be smaller. However, if the porosity is the same, shouldn't the number of pores increase as the size decreases? The number appears to be reduced in SEM images.
Round 2
Reviewer 2 Report
The current manuscript has been revised appropriately. Now readers can see the novelty of this manuscript, the position of this paper in the scientific field, and the rational strategy to prepare the hydrogen-permselective membrane using radiation-induced graft polymerization. The citations are also almost appropriate. Therefore, now the manuscript is suitable for publication.
Reviewer 3 Report
Dear authors
The authors have satisfactorily revised the manuscript. We believe that it is acceptable.